# Equation of State of Quark–Gluon Matter in the Clustering-of-Color-Sources Approach

Aditya Nath Mishra [1,2], Guy Paić [3], Carlos Vales Pajares [4,5], Rolf P. Scharenberg [6] and B. K. Srivastava [6,*]

1 University Centre For Research and Development (UCRD), Chandigarh University, Mohali 140413, India; aditya.nath.mishra@cern.ch
2 Department of Physics, University Institute of Sciences, Chandigarh University, Mohali 140413, India
3 Instituto de Ciencias Nucleares, Universidad Nacional Autonoma de Mexico, Apartado 70-543, Mexico; guy.paic@cern.ch
4 Departamento de Fisica de Particulas, Universidale de Santiago de Compostela, 15782 Santiago de Compostela, Spain; carlos.pajares@usc.es
5 Instituto Galego de Fisica de Atlas Enerxias (IGFAE), Universidale de Santiago de Compostela, 15782 Santiago de Compostela, Spain
6 Department of Physics and Astronomy, Purdue University, West Lafayette, IN 47907, USA; schrnbrg@purdue.edu
* Correspondence: brijesh@purdue.edu; Tel.: +1-765-494-4874

**Abstract:** In the first few microseconds after the Big Bang, the hot dense matter was in the form of quark–gluon plasma consisting of free quarks and gluons. By colliding heavy nuclei at RHIC and LHC at a velocity close to the speed of light, we were able to recreate primordial matter and observe that matter after expansion and cooling. In the present work, we have analyzed the transverse-momentum spectra of charged particles in high-multiplicity $pp$ collisions at LHC energies $\sqrt{s} = 5.02$ and $13$ TeV, published by the ALICE Collaboration, using the Color-String Percolation Model. For heavy ions, Pb–Pb at $\sqrt{s_{NN}} = 2.76$ and $5.02$ TeV along with Xe–Xe at $\sqrt{s_{NN}} = 5.44$ TeV have been analyzed. The initial temperature was extracted both in low- and high-multiplicity events in $pp$ collisions. For $A - A$ collisions, the temperature was obtained as a function of centrality. A universal scaling in the temperature from $pp$ and $A - A$ collisions was obtained when multiplicity was scaled by the transverse interaction area. For the higher-multiplicity events in $pp$ collisions at $\sqrt{s} = 5.02$ and $13$ TeV, the initial temperature was above the universal hadronization temperature and was consistent with the creation of deconfined matter. From the measured energy density $\varepsilon$ and the temperature, the dimensionless quantity $\varepsilon/T^4$ was obtained, to obtain the degree of freedom of the deconfined matter.

**Keywords:** QGP; deconfinement

## 1. Introduction

The Quantum Chromodynamics (QCD) phase diagram is closely related to the history of the universe and can be probed by heavy-ion collisions. Of particular interest in the heavy-ion collision experiments are the details of the deconfinement and chiral transitions that determine the QCD phase diagram. One of the main challenges in this field is to simultaneously determine the initial temperature and the energy density of the matter produced in a collision and, hence, the number of thermodynamic Degrees Of Freedom (DOF) [1].

Several interesting features related to Quark–Gluon Plasma (QGP) formation—e.g., long-range rapidity correlations, the so-called "ridge", elliptic flow, and strangeness enhancement seen in heavy-ion collisions—are also observed in high-multiplicity $pp$ collisions at LHC energies [2–4].

The objective of the present work was to obtain thermodynamic properties such as initial temperature and the DOF in $pp$, Pb–Pb, and Xe–Xe collisions at LHC energies by analyzing the published ALICE data on the transverse-momentum spectra of charged

hadrons [5–8], using the Color-String Percolation Model (CSPM) [9]. This required the measurement of the initial thermalized temperature and the energy density at time ~1 fm/c of the hot matter produced in these high-energy hadron–hadron and nucleon–nucleon collisions. A letter was published, using the limited sets of data [10].

This approach has been successfully used to describe the initial stages in the soft region in high-energy nucleus–nucleus and nucleon–nucleon collisions [9–11]. The CSPM is, in fact, different from the hydrodynamics picture and is more in line with other studies where the interaction among strings [12–14] or the domain color structure [15,16] is taken into account. Lattice-Quantum-ChromoDynamics-(LQCD) simulations indicate that the non-perturbative region of hot QCD matter extends up to a temperature of 400 MeV, well above the universal hadronization temperature [17].

This paper is organized as follows: Section 2 describes the phenomenology of the Color-String Percolation Model. The measurement of color suppression factor $F(\xi)$ and its relation to temperature are presented in Sections 3 and 4. Sections 5–8 describe thermalization, energy density, and degrees of freedom.

## 2. The Color-String Percolation Model

Multiparticle production is currently described in terms of color strings stretched between the projectile and the target, which decay into new strings through color-neutral $q - \bar{q}$ pairs production and subsequently hadronize to produce the observed hadrons [9]. Color strings may be viewed as small areas in the transverse plane filled with color fields created by colliding partons. In terms of gluon color fields, they can be considered as the color flux tubes stretched between the colliding partons.

With the growing energy and size of the colliding system, the number of strings grows, and they start to overlap, forming clusters in the transverse plane, in a way very much similar to the disks in two-dimensional-(2D) percolation theory, as shown in Figure 1 [18,19]. At a certain critical density, $\xi_c = 1.2$, a macroscopic cluster appears (connected) that marks the percolation-phase transition [18].

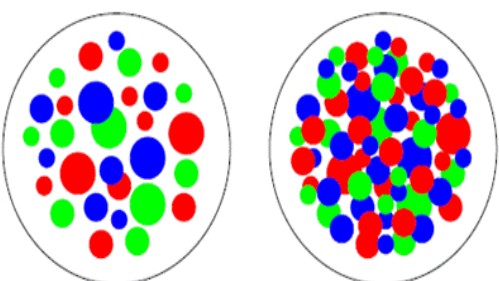

**Figure 1.** Partonic cluster structure in the transverse-collision plane at low (**left**) and high (**right**) parton density [19].

The interaction between the strings occurs when they overlap, and the general result—due to the SU(3) random summation of charges—is a reduction in multiplicity, an increase in the string tension, and, hence, an increase in the average transverse momentum squared, $\langle p_T^2 \rangle$. This is the Color-String Percolation Model [9].

The particle creation takes place by the Schwinger-$QED_2$ mechanism and is due to the color string breaking. The initiating colliding quarks and anti-quarks interact, to form a large number of color strings. The non-perturbative Schwinger-particle-creating mechanism in quantum electrodynamics $QED_2$, with massless fermions, was derived in an exact gauge-invariant calculation [20]. $QED_2$ contains a single space-and-time coordinate. Confinement, charge screening, asymptotic freedom, and the existence of a neutral bound-state boson in $QED_2$ closely models $QCD$. When string color fields are present, the Schwinger-$QED_2$-string-breaking mechanism lifts color-neutral $q\bar{q}$ pairs from the vacuum [21]. String breaking proceeds in an iterative way until they come to objects with masses comparable to hadron masses, which can be identified with observable hadrons by combining the

produced flavor with statistical weights [20,21]. The Schwinger mechanism has also been used in the decay of the color flux tubes produced by the quark–gluon plasma for modeling the initial stages in heavy-ion collisions [22,23].

We assume that a cluster of $n$ strings that occupies an area of $S_n$ behaves as a single color source with a higher color field $\vec{Q}_n$ corresponding to the vectorial sum of the color charges of each individual string $\vec{Q}_1$. The resulting color field covers the area of the cluster. As $\vec{Q}_n = \sum_1^n \vec{Q}_1$, and as the individual string colors may be oriented in an arbitrary manner, with respect to one another, the average $\vec{Q}_{1i}\vec{Q}_{1j}$ is zero and $\vec{Q}_n^2 = n\vec{Q}_1^2$. This results in the suppression of the multiplicity and the enhancement of $\langle p_T^2 \rangle$.

Knowing the color charge $\vec{Q}_n$, one can obtain the multiplicity $\mu_n$ and the mean transverse momentum squared $\langle p_T^2 \rangle_n$ of the particles produced by a cluster of $n$ strings [9]:

$$\mu_n = \sqrt{\frac{nS_n}{S_1}}\mu_1; \quad \langle p_T^2 \rangle_n = \sqrt{\frac{nS_1}{S_n}}\langle p_T^2 \rangle_1, \tag{1}$$

where $\mu_1$ and $\langle p_T^2 \rangle_1$ are the mean multiplicity and $\langle p_T^2 \rangle$ of particles produced from a single string with a transverse area $S_1 = \pi r_0^2$, where $r_0$ is the string radius. For strings just touching one another, $S_n = nS_1$, $\mu_n = n\mu_1$, and $\langle p_T^2 \rangle_n = \langle p_T^2 \rangle_1$. When strings fully overlap, $S_n = S_1$ and, therefore, $\mu_n = \sqrt{n}\mu_1$ and $\langle p_T^2 \rangle_n = \sqrt{n}\langle p_T^2 \rangle_1$, so that the multiplicity is maximally suppressed and the $\langle p_T^2 \rangle_n$ is maximally enhanced. This implies a simple relation between the multiplicity and transverse momentum $\mu_n\langle p_T^2 \rangle_n = n\mu_1\langle p_T^2 \rangle_1$, which means conservation of the total transverse momentum produced.

In the thermodynamic limit, one obtains an analytic expression [9],

$$\langle \frac{nS_1}{S_n} \rangle = \frac{\xi}{1 - e^{-\xi}} \equiv \frac{1}{F(\xi)^2}, \tag{2}$$

where $F(\xi)$ is the color-suppression factor and $\xi = \frac{N_s S_1}{S_N}$ is the percolation density parameter.

Equation (1) can be written as $\mu_n = nF(\xi)\mu_1$ and $\langle p_T^2 \rangle_n = \langle p_T^2 \rangle_1 / F(\xi)$. The critical cluster which spans $S_N$ appears for $\xi_c \geq 1.2$ [18,19]. $F(\xi)$ reduces the multiplicity and increases the transverse momentum of particles.

It is worth noting that the CSPM is a saturation model similar to Color Glass Condensate (CGC), where $\langle p_T^2 \rangle_1 / F(\xi)$ plays the same role as the saturation momentum scale $Q_s^2$ in the CGC model [24]. Saturation results from the overcrowding in the impact parameter of low $x$ partons of a boosted hadron or nucleus, leading to the appearance of a scale, $Q_s^2$. This is the basic idea of CGC. For example, the particle density in the CSPM is given by

$$\frac{dn}{dy} \sim (1 - e^{-\xi})^{1/2} N_{part}. \tag{3}$$

In CGC, particle density is related to the coupling constant $\alpha_s(Q_s^2)$:

$$\frac{dn}{dy} \sim \frac{1}{\alpha_s(Q_s^2)} N_{part}. \tag{4}$$

In both cases, particle density increases with the number of participants $N_{part}$. The correction to the $N_{part}$ scaling in the CGC is due to the occupation number given by $1/\alpha_s$, which gives rise to a $log(N_{part})$ dependence. In the CSPM, the correction is given by the factor $(1 - exp(-\xi))^{1/2}$ (Equation (3)), which is also a measure of the occupation, and, indeed, is the fraction of the collision area occupied by strings.

## 3. Determination of Color-Suppression Factor $F(\xi)$

In the present work, we have extracted the color-suppression factor $F(\xi)$ in high-multiplicity events in *pp* collisions using ALICE data from the transverse-momentum

spectra of charged particles at $\sqrt{s}$ = 5.02 and 13 TeV [5]. ALICE has obtained the transverse-momentum distribution for two multiplicity estimators that cover different pseudorapidity regions. The estimators are based on either the total charge deposited in the forward-detector (covering the pseudorapidity regions $2.8 < \eta < 5.1$ and $-3.7 < \eta < -1.7$) V0M or on the number of tracks in the pseudorapidity region $|\eta| < 0.8$ $N_{SPDtracklets}$. Table 1 shows the event-multiplicity classes based on V0M and $N_{SPDtracklets}$ for $pp$ collisions at $\sqrt{s} = 5.02$ and 13 TeV [5]. Table 1 also shows the various multiplicity classes based on the V0M estimator for $pp$ at $\sqrt{s} = 7$ TeV [6].

**Table 1.** Event-multiplicity classes based on the number of tracklets ($N_{SPDtracklets}$) within $|\eta| < 0.8$ for $pp$ collisions. For V0M, it covers the region $2.8 < \eta < 5.1$ and $-3.7 < \eta < -1.7$. In both cases, $< dN_{ch}/d\eta >$ is given in the region $|\eta| < 0.8$ [5–8].

| System | *pp* 13 TeV | *pp* 13 TeV | *pp* 5.02 TeV | *pp* 5.02 TeV | *pp* 7 TeV |
|---|---|---|---|---|---|
| | $N_{SPDtracklets}$ | V0M | $N_{SPDtracklets}$ | V0M | V0M |
| **Multiplicity Class** | $< dN_{ch}/d\eta >$ | $< dN_{ch}/d\eta >$ | $< dN_{ch}/d\eta >$ | $< dN_{ch}/d\eta >$ | $< dN_{ch}/d\eta >$ |
| I | 54.1 | 26.6 | - | 19.2 | 21.3 |
| II | 44.6 | 20.5 | 34.6 | 15.1 | 16.5 |
| III | 38.9 | 16.7 | 29.9 | 12.4 | 13.5 |
| IV | 34.1 | 14.3 | 26.2 | 10.7 | 11.5 |
| V | 29.3 | 12.6 | 22.4 | 9.47 | 10.1 |
| VI | 24.5 | 10.6 | 18.5 | 8.04 | 8.45 |
| VII | 19.5 | 8.46 | 14.6 | 6.56 | 6.72 |
| VIII | 14.4 | 6.82 | 10.6 | 5.39 | 5.40 |
| IX | 9.03 | 4.94 | 6.58 | 4.05 | 3.90 |
| X | 2.91 | 2.54 | 2.21 | 2.07 | 2.26 |

Figure 2 shows the transverse-momentum spectra for two multiplicity cuts at $\sqrt{s} = 13$ TeV in $pp$ collisions and Pb–Pb at $\sqrt{s_{NN}} = 5.02$ TeV for 0–5% centrality. The spectra become harder for higher-multiplicity cuts. This is due to the fact that high-string-density color sources are created in the higher-multiplicity events. To evaluate the initial value of $F(\xi)$ from the data in high-multiplicity events in $pp$ collisions, a parameterization of the experimental data of $p_T$ distribution in low-energy $pp$ collisions at $\sqrt{s} = 200$ GeV was used [9]. The charged-particle spectrum is described by a power law [9],

$$d^2 N_c / dp_T^2 = a / (p_0 + p_T)^\alpha, \tag{5}$$

where $a$ is the normalization factor, and where $p_0$ and $\alpha$ are fitting parameters with $p_0 = 1.98$ and $\alpha = 12.87$. This parameterization is used in high-multiplicity $pp$ collisions to take into account the interactions of the strings [9]. The color-suppression factor $F(\xi)$ encodes the effects of the interaction among the strings once they overlap. The parameter $p_0$ in Equation (5) is for independent strings and becomes modified:

$$p_0 \rightarrow p_0 \left( \frac{\langle nS_1/S_n \rangle_{pp}^{mult}}{\langle nS_1/S_n \rangle_{pp}} \right)^{1/4}, \tag{6}$$

$$\frac{d^2 N_c}{dp_T^2} = \frac{a}{(p_0 \sqrt{F(\xi)_{pp}/F(\xi)_{pp}^{mult}} + p_T)^\alpha}, \tag{7}$$

where $F(\xi)_{pp}^{mult}$ is the multiplicity-dependent color-suppression factor. In $pp$ collisions, $F(\xi)_{pp} \sim 1$ at low energies, due to the low overlap probability. The spectra were fitted using Equation (7) in the softer sector, with $p_T$ in the range 0.12–1.0 GeV/c. In the thermodynamic limit, the color-suppression factor $F(\xi)$ is related to the percolation density parameter $\xi$ [9]:

$$F(\xi) = \sqrt{\frac{1 - e^{-\xi}}{\xi}}.$$ (8)

Figure 3 shows the extracted value of $F(\xi)$ as a function of $< dN_{ch}/d\eta >$ ($N_{tracks}$) from the ALICE experiment for $\sqrt{s}$ = 5.02 and 13 TeV using both estimator $SPD_{tracklets}$ and V0M. In the case of $\sqrt{s}$ = 7 TeV, only the V0M estimator results are shown. $N_{tracks}$ is the charged-particle multiplicity covering the pseudo-rapidity range $|\eta| < 0.8$ and $p_T$ = 0.15–20 GeV/c [5]. It is observed that for fixed-average charged-particle multiplicity, $F(\xi)$ has a similar value for all three energies. As $SPD_{tracklets}$ covers higher multiplicity, further studies are shown only with this estimator. Figure 4a,b show $F(\xi)$ and $\xi$ as a function of $< dN_{ch}/d\eta >$.

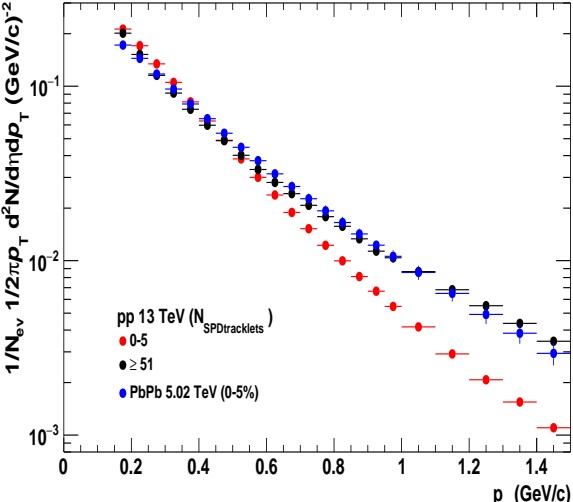

**Figure 2.** Transverse-momentum spectra of charged particles from ALICE experiment in *pp* collisions at $\sqrt{s}$ = 13 TeV for two different multiplicity cuts, $N_{SPDtracklets} \geq 51$ (purple solid circle) and $N_{SPDtracklets} < 5$ (red solid circle) [5]. Pb-Pb collision spectra at $\sqrt{s_{NN}}$ = 5.02 TeV for 0–5% centrality are shown as blue solid circles [7].

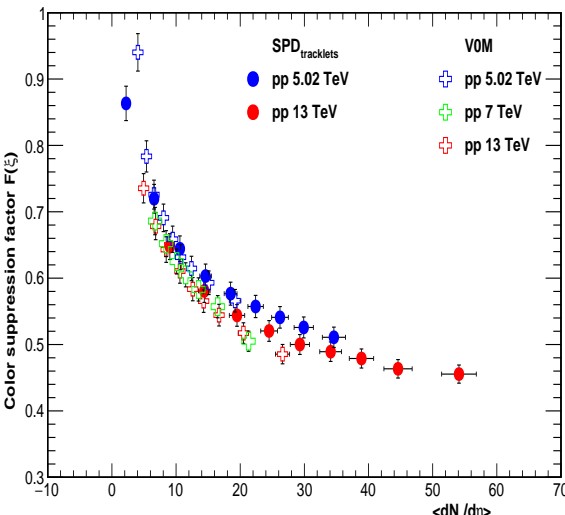

**Figure 3.** Color-suppression factor $F(\xi)$ in *pp* collisions vs. $< dN_{ch}/d\eta >$ for $SPD_{tracklets}$ and V0M event-multiplicity classes. For *pp* collisions at $\sqrt{s}$ = 7 TeV, only V0M event-multiplicity classes are available. The charged-particle multiplicity covering the kinematic range $|\eta| < 0.8$ is $< dN_{ch}/d\eta >$ and transverse momentum $p_T$ = 0.15–20 GeV/c [5].

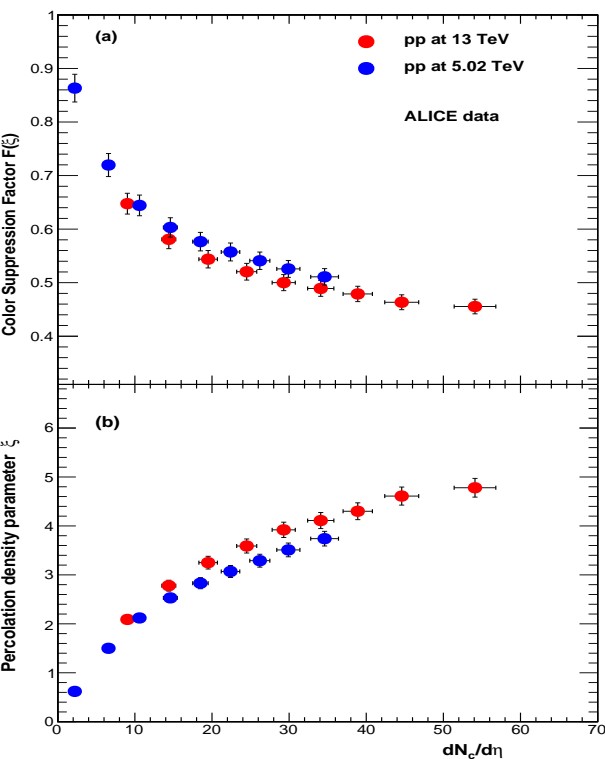

**Figure 4.** (**a**) Color-suppression factor $F(\xi)$. (**b**) String density $\xi$ in *pp* collisions vs. $dN_{ch}/d\eta$. The charged-particle multiplicity covering the kinematic range $|\eta| < 0.8$ is $dN_{ch}/d\eta$, and $p_T = 0.15$–20 GeV/c [5].

In cases of $A - A$ collisions, $\langle nS_1/S_n \rangle_{pp}^{mult}$ in Equations (6) and (7) is replaced by $\langle nS_1/S_n \rangle_{AA}$ and $F(\xi)_{AA}$, respectively. For Pb–Pb at $\sqrt{s_{NN}} = 2.76$ and 5.04 TeV and Xe–Xe at $\sqrt{s_{NN}} = 5.44$, TeV $F(\xi)$ has been extracted from the spectra at various centralities [7,8]. Figure 5a,b show $F(\xi)$ and $\xi$ for, respectively, Xe–Xe and Pb–Pb collisions.

To compare *pp* with the heavy-ions results, we need to normalize $N_{tracks}$ with the transverse area $S_\perp$ in *pp* and $A - A$ collisions. In *pp* collisions, $S_\perp = \pi R_{pp}^2$ has been computed in the IP-Glasma model, where $R_{pp}$ is the interaction radius [25]. This is based on an impact-parameter description of *pp* collisions, combined with an underlying description of particle production based on the theory of CGC [25]. The interaction radius $R_{pp}$ is approximately a linear function of the charged-particle multiplicity. In the IP-Glasma model, $R_{pp}$ is dependent on gluon multiplicity [25]:

$$R_{pp} = f_{pp}(dN_g/dy)^{1/3}; \tag{9}$$

$f_{pp} = (0.387 + 0.0335x + 0.274x^2 - 0.0542x^3)$ for $x < 3.4$ and $f_{pp} = 1.538$ for $x \geq 3.4$, where $x = (dN_g/dy)^{1/3}$. The gluon multiplicity $dN_g/dy$ is related to the number of tracks seen in the CMS experiment by

$$dN_g/dy \approx (3/2)\frac{1}{\Delta\eta}N_{track}, \tag{10}$$

where $\Delta\eta \sim 4.8$ units of pseudorapidity. The interaction cross section $S_\perp$ for *pp* collisions at $\sqrt{s} = 5.02$ and 13 TeV from ALICE [5] is shown in Figure 6. $S_\perp$ increases with the multiplicity, and for very high multiplicities it is approximately constant. In the case of A–A collisions, the nuclear overlap area was obtained using the Glauber model [26].

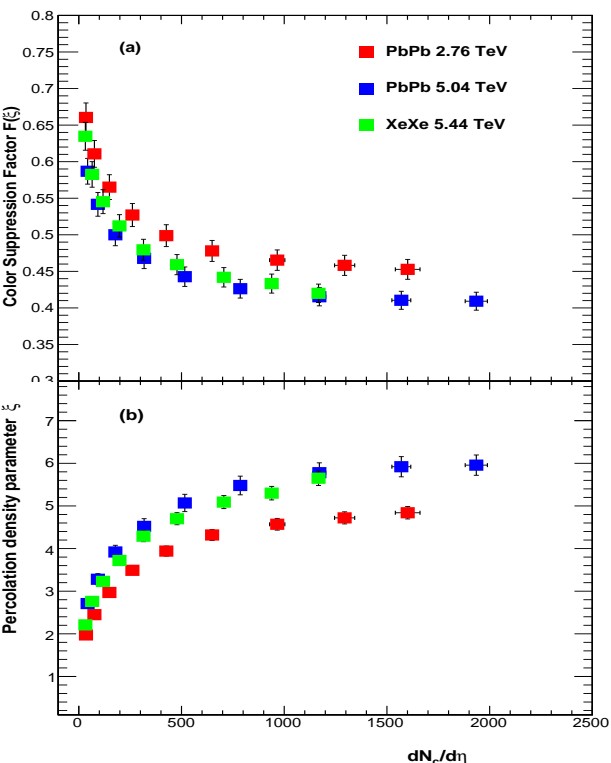

**Figure 5.** (**a**) Color-suppression factor $F(\xi)$. (**b**) String density $\xi$ in Pb–Pb, and Xe–Xe collisions vs. $dN_{ch}/d\eta$. The charged-particle multiplicity covering the kinematic range $|\eta| < 0.8$ is $dN_{ch}/d\eta$, and $p_T = 0.15$–20 GeV/c [7,8].

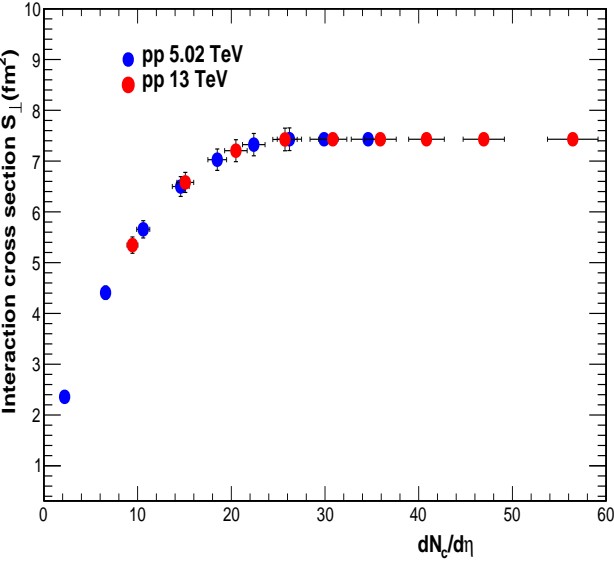

**Figure 6.** Interaction cross section $S_\perp$ vs. $dN_{ch}/d\eta$. $S_\perp$ is obtained using the IP-Glasma model [25].

Figure 7a shows $F(\xi)$ as a function of $dN_{ch}/d\eta$ scaled by transverse area $S_\perp$ for *pp*, Pb–Pb, and Xe–Xe collisions. Percolation density parameter $\xi$ is shown in Figure 7b. The results are also shown for Au–Au collisions at $\sqrt{s_{NN}} = 200$ GeV [9]. A universal scaling behavior is observed in hadron–hadron and nucleus–nucleus collisions.

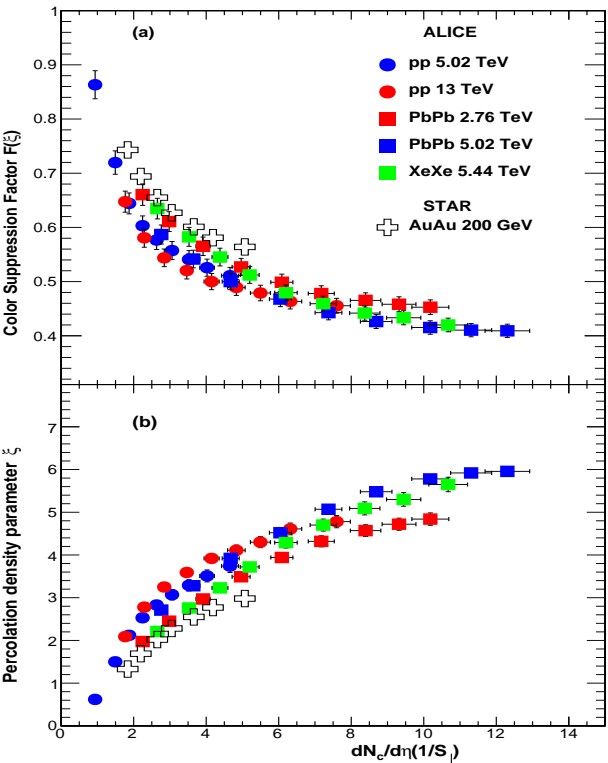

**Figure 7.** (**a**) Color-suppression factor $F(\xi)$ in *pp*, Pb–Pb, and Xe–Xe collisions vs. $dN_{ch}/d\eta$ scaled by the transverse area $S_\perp$. For *pp* collisions, $S_\perp$ is multiplicity dependent, as obtained from the IP-Glasma model [25]. In the case of Pb–Pb and Xe–Xe collisions, the nuclear overlap area was obtained using the Glauber model [26]. (**b**) String density parameter $\xi$. The results are also shown for Au–Au collisions at $\sqrt{s_{NN}}$ = 200 GeV [9].

## 4. Connection between $F(\xi)$ and Temperature

The connection between $F(\xi)$ and the temperature $T(\xi)$ involves the Schwinger Mechanism (SM) for particle production. The Schwinger distribution for massless particles is expressed in terms of $p_t^2$ [20,21]:

$$dn/dp_t^2 \sim \exp(-\pi p_t^2/x^2), \tag{11}$$

where the average value of the string tension is $\langle x^2 \rangle$. The tension of the macroscopic cluster fluctuates around its mean value because the chromo-electric field is not constant. The origin of the string fluctuation is related to the stochastic picture of the QCD vacuum. As the average value of the color field strength must vanish, it cannot be constant but changes randomly from point to point [27,28]. Such fluctuations lead to a Gaussian distribution of the string tension,

$$\frac{dn}{dp_t^2} \sim \sqrt{\frac{2}{<x^2>}} \int_0^\infty dx \exp(-\frac{x^2}{2<x^2>}) \exp(-\pi\frac{p_t^2}{x^2}), \tag{12}$$

which gives rise to thermal distribution [27],

$$\frac{dn}{dp_t^2} \sim \exp\left(-p_t\sqrt{\frac{2\pi}{\langle x^2 \rangle}}\right), \tag{13}$$

with $\langle x^2 \rangle = \pi\langle p_t^2 \rangle_1/F(\xi)$. The temperature is expressed as [9]

$$T(\xi) = \sqrt{\frac{\langle p_t^2 \rangle_1}{2F(\xi)}}. \tag{14}$$

The string-percolation density parameter $\xi$, which characterizes the percolation clusters, measures the initial temperature of the system. As this cluster covers most of the interaction area, the temperature becomes a global temperature determined by the string density. In this way, at $\xi_c = 1.2$ the connectivity-percolation transition at $T(\xi_c)$ models the thermal-deconfinement transition.

Figure 8 shows the temperature as a function of $dN_c/d\eta$ scaled by the interaction area for $pp$, Xe–Xe, and Pb–Pb collisions at LHC energies. Temperatures from both hadron–hadron and nucleus–nucleus collisions fall on a universal curve when multiplicity is scaled by the transverse interaction area. The horizontal line at temperature T $\sim$ 167.7 MeV is the universal hadronization temperature obtained from the systematic comparison of the statistical-thermal-model parameterization of hadron abundances measured in high-energy $e^+e^-$, $pp$, and $A - A$ collisions [29]. In Figure 8, for $pp$ collisions at $\sqrt{s} = 5.02$ and 13 TeV, higher-multiplicity cuts show temperatures above the hadronization temperature and similar to those observed in Au–Au collisions at $\sqrt{s_{NN}} = 200$ GeV [9].

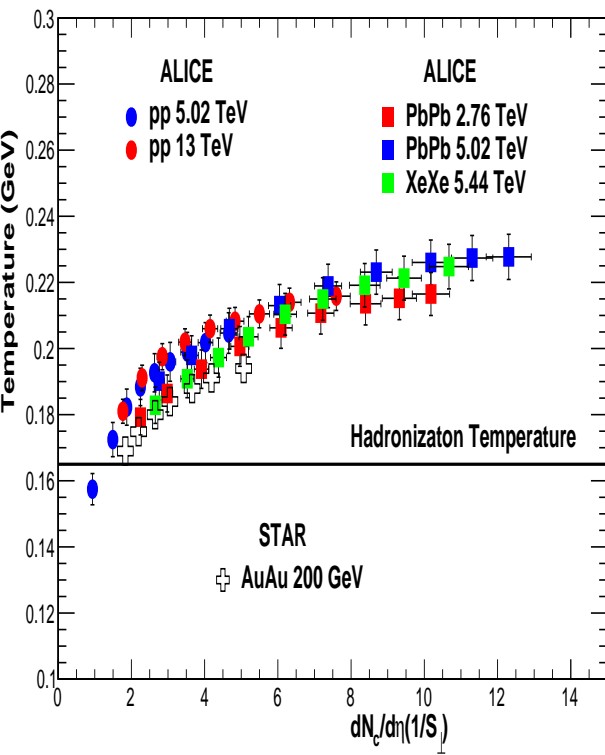

**Figure 8.** Temperature vs. $dN_{ch}/d\eta$ scaled by $S_\perp$ from $pp$, Pb–Pb, and Xe–Xe collisions. The horizontal line at temperature T $\sim$ 165 MeV is the universal hadronization temperature [29].

It is observed that the temperature rises slowly at higher $dN_c/d\eta$ values. This behavior is related to the overlap area of strings, which is given by $(1 - e^{-\xi})$. Above $\xi \sim 4$ there is complete overlap and the temperature rises slowly, as $\xi^{1/4}$. Temperature and $\xi$ are shown in Figure 9a,b as a function of the number of participants $N_{part}$. It is observed that the string density $\xi$ is collision-energy dependent for the same number of participants.

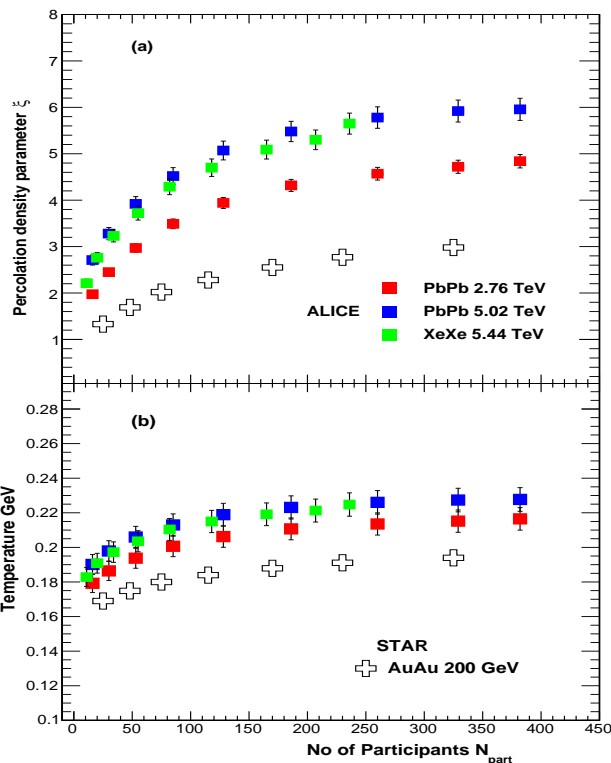

**Figure 9.** (**a**) Percolation density parameter $\xi$. (**b**) Temperature as a function of the number of participants $N_{part}$ in Pb–Pb and Xe–Xe collisions along with Au–Au at $\sqrt{s_{NN}}$ = 200 GeV [9].

## 5. Thermalization

In cases of $A - A$ collisions, it is observed that the average transverse momentum is twice the temperature $< p_T > \sim$ 2T [9]. This shows that the charged-particle transverse-momentum spectrum is exponentially distributed and that the inverse-slope parameter is the thermalized temperature [9]. Similar behavior is observed in $pp$ collisions, as shown in Figure 10 along with Pb–Pb and Xe–Xe results.

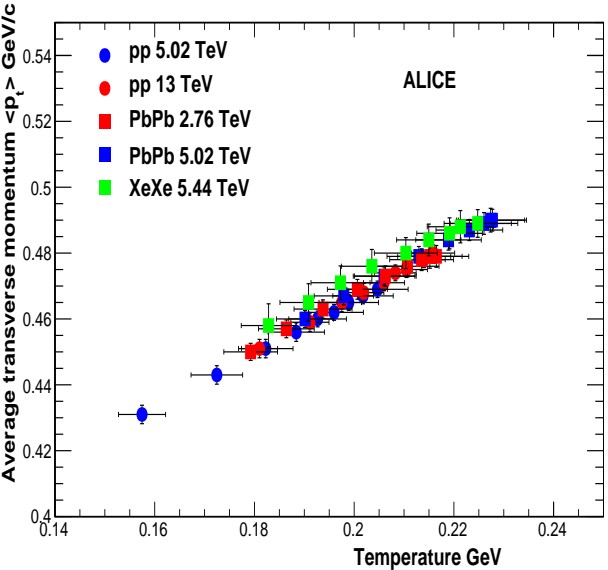

**Figure 10.** Average transverse momentum vs. temperature.

The thermalization in *pp* and $A - A$ collisions can also occur through the existence of an event horizon caused by a rapid deceleration of the colliding nuclei [30]. The thermalization is due to the Hawking–Unruh effect [30–32]. It is well known that black holes evaporate by quantum-pair production and behave as if they have an effective temperature of

$$T_H = \frac{1}{8\pi GM},$$

(15)

where 1/4GM is the acceleration of gravity at the surface of a black hole of mass M. The rate of pair production in the gravitational background of the black hole can be evaluated by considering the tunneling through the event horizon. Unruh showed that a similar effect arises in a uniformly accelerated frame, where an observer detects thermal radiation with the temperature T = a/2, where *a* is the acceleration. Similarly, in hadronic interactions the probability of producing states of masses M due to the chromo-electric field E and the color charge is given by the Schwinger Mechanism [20],

$$W_M \sim \exp(\frac{-\pi M^2}{gE}) \sim \exp(-M/T),$$

(16)

which is similar to the Boltzmann weight in a heat bath with an effective temperature

$$T = \frac{a}{2\pi}, a = \frac{2gE}{M}.$$

(17)

In CSPM, the strong color field inside the large cluster produces de-acceleration of the primary $q\bar{q}$ pair, which can be seen as thermal temperature by means of the Hawking–Unruh effect. This implies that the radiation temperature is determined by the transverse extension of the color flux tube/cluster, in terms of the string tension [30,33]:

$$T = \sqrt{\frac{\sigma}{2\pi}},$$

(18)

where $\sigma$ is string tension. This string tension, referred to as the tension of a cluster of strings, can be written in terms of the color-suppression factor, in such a way that Equation (18) is the same as Equation (14). This is not surprising because both the Hawking–Unruh effect and Equation (14) are based on the Schwinger Mechanism.

The string percolation density parameter $\xi$ that characterizes the percolation clusters measures the initial temperature of the system. As this cluster covers most of the interaction area, this temperature becomes a global temperature determined by the string density. In this way, at $\xi_c$ = 1.2 the connectivity-percolation transition at $T(\xi_c)$ models the thermal-deconfinement transition. The temperature obtained using Equation (14) was ∼193.6 MeV for Au–Au collisions at $\sqrt{s_{NN}}$ = 200 GeV, in reasonable agreement with $T_i = 221 \pm 19^{stat} \pm 19^{sys}$ MeV from the enhanced-direct-photon experiment measured by the PHENIX Collaboration [34]. For Pb–Pb collisions at $\sqrt{s_{NN}}$ = 2.76 TeV, the temperature is ∼262.2 MeV for 0–5% centrality, which is expected to be ∼35% higher than the temperature from Au–Au collisions [9]. A recent summary of the results from Pb–Pb collisions at the LHC has mentioned that the initial temperature increases by at least 30%, as compared to the top RHIC energy [35]. The direct photon measurements from ALICE give the temperature of $T_i = 304 \pm 51$ MeV [36]. The agreement with the measured temperature shows that the temperature obtained using Equation (14) can be termed as the initial temperature of the percolation cluster.

## 6. Energy Density

The QGP according to the CSPM is born in local thermal equilibrium because the temperature is determined at the string level. After the initial temperature $T > T_c$ the CSPM perfect fluid may expand according to Bjorken boost-invariant 1D hydrodynamics [37]

$$\varepsilon = \frac{3}{2} \frac{\frac{dN_c}{dy} \langle m_T \rangle}{S_n \tau_{pro}},$$ (19)

where $\varepsilon$ is the energy density, $S_n$ is the nuclear-overlap area, $m_T$ is the transverse mass, and $\tau_{pro}$ is the production time for a boson (gluon) [21]. $\tau_{pro}$ for a boson (gluon) is given by [20]

$$\tau_{pro} = \frac{2.405\hbar}{\langle m_T \rangle}.$$ (20)

Above the critical temperature only massless particles are present in the CSPM. In Figure 11, energy density as a function of string density is shown for $pp$ and $A - A$ collisions at LHC energies. The results for $pp$ at $\sqrt{s}$ = 13 TeV and Pb–Pb at $\sqrt{s_{NN}}$ = 2.76 and 5.02 TeV are from our earlier publication [10]. Figure 11 also has results from Au–Au at $\sqrt{s_{NN}}$ = 200 GeV [9]. We observe a slow rise of $\varepsilon$ for low values of $\xi$ followed by a faster rise later. It is found that $\varepsilon$ is proportional to $\xi$ in the range $1.2 < \xi < 5.0$. Above $\xi \sim 5$, the energy density $\varepsilon$ rises faster compared to $\xi < 5$. A possible explanation for the sharp rise in the energy density could be that at such a high degree of overlapping the gluons are seen naked without interaction and, thus, the coherence of the color fields of the overlapping strings is lost; thus, the independence of the strings is recovered. This means that instead of a dependence on $N_{part}$ it would be $N_{part}^{4/3}$.

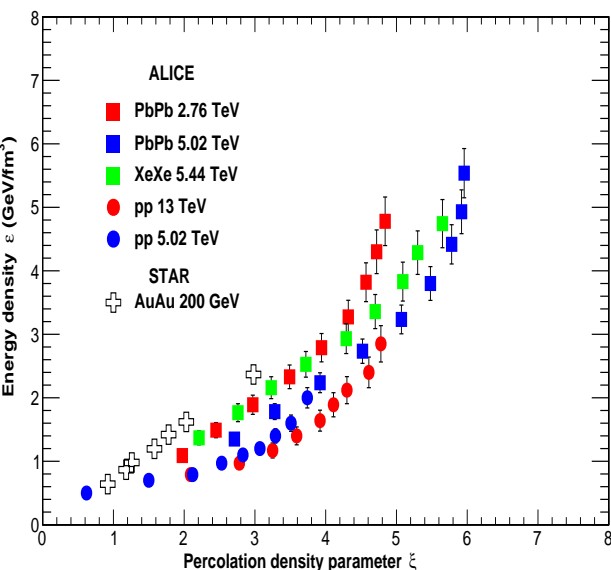

**Figure 11.** Energy density ($\epsilon$) as a function of the percolation density parameter ($\xi$) for $pp$ collisions at $\sqrt{s}$ = 5.02 and 13 TeV, Pb–Pb collisions at $\sqrt{s_{NN}}$ = 2.76 and 5.02 TeV, and Xe–Xe collisions at $\sqrt{s_{NN}}$ = 5.44 TeV. The results from Au–Au collisions at $\sqrt{s_{NN}}$ = 200 GeV are also shown [9].

It is worth mentioning that the CSPM is only used to encode the dependence on the centrality and energy of the temperature T given by the $p_T$ spectrum. As far as this evaluation can be done directly from the $p_T$ spectrum independently of the CSPM, our result can be seen independent of the model used.

Energy density has been obtained in a lattice set-up of (2 + 1)-flavor QCD, using the Highly-Improved-Staggered-Quark-(HISQ) action and the tree-level improved gauge action [38,39]. Figure 12 shows dimensionless quantity $\varepsilon/T^4$ as a function of temperature

both from the CSPM and LQCD. It is observed that the CSPM results show a similar trend to the LQCD results up to the temperature of T $\sim$ 210 MeV. Beyond this temperature, the $\varepsilon/T^4$ in the CSPM rises much faster and reaches the ideal gas value of $\varepsilon/T^4 \sim$ 16 at T $\sim$ 230 MeV. In this region, there is a strong screening, due to the large degree of overlapping of the strings, producing a faster approach to the quark–gluon gas limit.

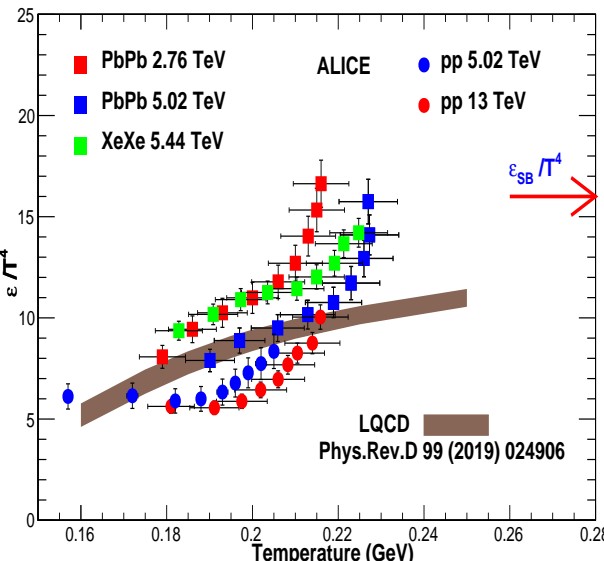

**Figure 12.** Dimensionless quantity $\varepsilon/T^4$ as a function of temperature from the CSPM and LQCD calculation from HotQCD Collaboration [38]. Stefan–Boltzmann limit for ideal gas of quarks and gluons is also shown at $\varepsilon_{SB}/T^4$. The CSPM values at higher temperature T > 200 MeV are obtained by extrapolating from lower temperature.

### 7. CSPM and Two Temperatures

In the CSPM, we can associate two scales and the corresponding temperatures to the two different transitions. One is the string density required to have a large cluster of strings crossing the surface of the collision. This happens at the critical percolation density:

$$\frac{N}{S_\perp} = \frac{\xi_c}{\pi r_0^2}.$$ (21)

At this critical percolation density a cluster of strings is extended over most of the collision's surface and, thus, the color field. In this way, the quarks and gluons can be considered confined. The second temperature could be associated with the restoration of chiral symmetry, considering the argument: from the point of view of statistical mechanics the formed deconfined medium can form a fluid, such as has been observed, only if the constituents—the color sources—have a hard core [40]. The size of this hard core $h$ provides us with a new scale. On the other hand, as the area covered by the strings is $(1 - exp(-\xi))S_\perp$, the mean distance between strings $d$ is given by

$$d = \left[\frac{N}{(1 - exp(-\xi))S_\perp}\right]^{-1/2} = F(\xi)\sqrt{\pi}r_0.$$ (22)

For small $\xi$, $d > r_0$: for example, at the critical percolation density $\xi_c$ = 1.2, d = 1.34$r_0$. This means that overlapping between strings is very peripheral, covering only the edges (corona) of the strings. This corresponds to the hadronization temperature of $\sim$166 MeV. In order to penetrate the core, the overlapping should be larger, in such a way that $d = 2h < r_0$. For reasonable values of $h \sim$ 0.4, 0.5 ($\sim$half of the ratios of the string), we have $F(\xi) \sim$ 0.45–0.50 corresponding to the temperature of $\sim$210–220 MeV. This is the temperature at which our result starts deviating from LQCD. The strong overlapping of strings penetrating the hard

core means that we are deep inside the color cloud surrounding the source, in such a way that the source appears undressed, i.e., total deconfinement.

## 8. Degrees of Freedom

In the case of a quark–gluon system in thermal equilibrium at a high temperature, the quarks and gluons are idealized to be non-interacting and massless and there is no net baryon number. The number of quarks and antiquarks are equal [19,21]. Adding the contribution from quarks, antiquarks, and gluons, the total energy density of an ideal QGP is given by [21]

$$\varepsilon = g_{total} \frac{\pi^2}{30} T^4, \tag{23}$$

where

$$g_{total} = [g_g + \frac{7}{8} \times (g_q + g_{\bar{q}}), \tag{24}$$

where $g_g$, $g_q$, and $\bar{q}$ are, respectively, the degeneracy number of the gluons, the quarks, and the antiquarks. There are eight gluons and two possible polarizations, giving $g_g$ = 16 DOF. The degeneracy numbers $g_q$ and $\bar{q}$ are

$$g_q = \bar{q} = N_c N_s N_f, \tag{25}$$

where $N_c$ = 3 is the number of colors, $N_s$ = 2 is the number of spins, and $N_f$ = 2 or 3 is the number of flavors. Thus, the energy densities obtained in full QCD with different numbers of quark flavor using the above equations are given by

$$\varepsilon/T^4 \simeq \frac{37}{30} \pi^2 \simeq 12, N_f = 2. \tag{26}$$

$$\varepsilon/T^4 \simeq \frac{47.5}{30} \pi^2 \simeq 16, N_f = 3. \tag{27}$$

At T $\sim$ 210 MeV, $\varepsilon/T^4 \sim$ 11, which corresponds to $\sim$33 DOF, while at T $\sim$ 230 MeV there are $\sim$47 DOF. In Figure 12, the Stefan–Boltzmann limit is also shown at $\varepsilon_{SB}/T^4$, which corresponds to $\sim$48 DOF. It is observed that Pb–Pb at $\sqrt{s_{NN}}$ = 2.76 TeV has similar features as seen at 5.02 TeV. For Xe–Xe collisions at $\sqrt{s_{NN}}$ = 5.44 TeV, $\sim$44 DOF are obtained. In *pp* collisions at $\sqrt{s}$ = 13 TeV only $\sim$ 33 DOF are reached. Our results agree with the conclusions obtained by studying the trace anomaly in a quasi-particle gluonic model [41,42]. In this model, the DOF of the free gluons are also obtained for T $\simeq$ 1.3T$_c$ (T$_c \approx$ 165 MeV).

## 9. Discussion

The most pressing question to be addressed is the appearance of a transition in the CSPM at T $\sim$ 210 MeV above the deconfinement transition. It is quite possible that temperature $\sim$ 210 MeV is the chiral-symmetry-restoration temperature. In that case, the deconfinement temperature occurs at a lower temperature than the chiral-symmetry-restoration temperature. However, the LQCD calculation of the equation of state in 2 + 1-flavor QCD at a finite temperature finds that deconfinement and chiral-symmetry restoration occur in the same temperature range, i.e., in a region of the QCD phase transition where the transition is not a true phase transition but is, rather, a rapid crossover [43]. Chiral condensate using physical quark masses has also been obtained in LQCD. At temperatures above 200 MeV, subtracted chiral condensate drops to zero. This temperature is close to the CSPM first temperature [44].

It has been suggested that chiral-symmetry restoration occurs either together with or after color deconfinement [19]. Deconfinement is the transition from a state of color-neutral hadrons to one of the colored quarks. Chiral-symmetry restoration is the transition from a state of massive "dressed" constituent quarks to one of massless current quarks. These two phenomena need not coincide [19,45].

A new phase in QCD also has been proposed, studying the Dirac operator [46]. While confining chromo-electric interaction is distributed among all modes of Dirac operator, chromo-magnetic interaction is located predominantly in the near-zero modes. Above $T \sim 155$ MeV, the near-zero modes are suppressed but not the rest of the modes, surviving the chromo-electric interaction, which is suppressed at higher temperatures [46]. Following this, it has been conjectured that the intermediate temperature regime is "stringy fluid", where chiral symmetry is stored [47].

Recently, using lattice simulations, in addition to the standard crossover-phase transition at $T \sim 155$ MeV, the existence of a new infrared-phase transition at temperature T, $200 < T_{IR} < 250$ MeV has been pointed out. In this phase, asymptotic freedom works and, therefore, there is no interaction. In between these two temperatures, there is possible coexistence of the short- and long-distance scales [48,49], which supports the present observation in our work.

Chiral symmetry restoration also has been studied by an accelerating observer considering the Unruh effect. As a result, it is shown that chiral symmetry is restored for uniformly accelerated observers with acceleration a larger than the critical value $a_c = 4\pi f_\pi$, with $f_\pi$ being the pion decay constant [50].

An alternative explanation of the results found here is related to the strong magnetic field, which is produced in pp collisions, and which is even stronger in $A - A$ collisions. The lattice studies of the QCD-chiral phase transition with three flavors in a background magnetic field show that chiral condensate and, thus, the temperature of this phase transition always increase with the magnetic field. The transition instead of a crossover becomes a first-order phase transition. As the magnetic field is higher in Pb–Pb than in $pp$, a higher temperature is expected for heavy-ion collisions [51]. This is in agreement with Figure 12. We plan to continue our work for narrower centrality bins. In the present work, we have used 0–5% centrality. We expect that $\varepsilon/T^4$ with a narrower bin in centrality, e.g., 0–2%, will not exceed the ideal gas of the quark and gluon limit.

## 10. Summary and Conclusions

We have used the CSPM to explore the initial stage in $pp$, Xe–Xe, and Pb–Pb collisions at LHC energies and we have determined the thermalized initial temperature of the hot nuclear matter at an initial time $\sim 1$ fm/c. For the first time, both the temperature and the energy density of the hot nuclear matter have been obtained from the measured charged-particle spectra using ALICE data for $pp$ collisions at $\sqrt{s} = 5.02$ and 13 TeV. In $A - A$ collisions, Pb–Pb at $\sqrt{s_{NN}} = 5.02$ and 5.44 TeV along with Xe–Xe at $\sqrt{s_{NN}} = 5.44$ TeV have been used. A universal scaling in the temperature was obtained for both $pp$ and $A - A$ (Figure 8), which was well above the universal hadronization temperature, indicating that the matter created was in the deconfined phase. Thermalization in both $pp$ and $A - A$ was reached through the stochastic process (Hawking–Unruh) rather than the kinetic approach. The dimensionless quantity $\varepsilon/T^4$ was evaluated, to obtain the DOF of the deconfined phase. We observed two features hitherto not reported: the existence of two temperature ranges in the behavior of the $A - A$ system DOF and a clear departure from the LQCD results regarding the maximum number of DOF, which reached values in agreement with the Stephan Boltzmann limit for an ideal gas of quarks and gluons. In the case of $pp$ collisions at $\sqrt{s} = 5.02$, we reached only $\varepsilon/T^4 \sim 8$, corresponding to $\sim 24$ DOF, while at $\sqrt{s} = 13$ TeV, $\sim 33$ DOF was obtained. It is worth noting the importance of our main result: namely, the departure of $\varepsilon/T^4$ from the LQCD result, starting at $T \sim 210$–220 MeV. As the energy density was obtained directly from the ALICE data on $dn/d\eta$ and $p_T$ distributions and as the temperature was determined by the mean $p_T$, which was obtained directly from the data, the result can be considered as an experimental result.

It has been argued that QCD could lead to a three-phase structure, as a function of the temperature. In such a scenario, color deconfinement would result in a plasma of massive "dressed" quarks; the only role of gluons in this state would be to dynamically generate the effective quark mass. The effective DOF in the resulting quark plasma, thus, are just those

of massive quarks. At still higher temperature, this gluonic dressing of quarks would then evaporate, leading to total deconfinement.

**Author Contributions:** Formal analysis, A.N.M.; writing—original draft preparation, B.K.S.; writing—review and editing, G.P. and C.V.P.; supervision, R.P.S. All authors have read and agreed to the published version of the manuscript.

**Funding:** G.P.'s work was supported by a grant from CONACYT under grant A1-S-22917 and CF-2042. C.V.P. gives thanks for the grant from Maria de Maeztu Unit of Excellence MDM-2016-0682 of Spain, the support of Xunta de Galicia under the project ED431C 2017, and project PID 2020-119632GB-IOO of the Ministerio de Ciencia e Innovacion of Spain and FEDER.

**Data Availability Statement:** Data are contained within the article.

**Acknowledgments:** A. N. Mishra acknowledges support from UCRD, Chandigarh University for the facilities provided to carry out the research.

**Conflicts of Interest:** The authors declare no conflict of interest.

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
