# Peer review of "Equation of State of Quark–Gluon Matter in the Clustering-of-Color-Sources Approach"

_universe, doi:10.3390/universe10020055_

Round 1
Reviewer 1 Report
Comments and Suggestions for Authors
In a comprehensive review of the paper titled "Equation of State of Quark Gluon Matter in the Clustering of Color Sources Approach" authored by A. N. Mishra et al., the study uses the Color String Percolation Model (CSPM) to investigate early-stage matter formation in high-energy collisions. Analyzing ALICE data for various collisions, identifies a universal temperature scaling, indicating the creation of deconfined matter. Deviations from theoretical predictions above 210-220 MeV suggest discrepancies in degrees of freedom, possibly signifying a transition in the quark-gluon plasma's nature and emphasizing the role of gluonic dressing in the quark plasma's effective degrees of freedom.
Provide a clear explanation of the methodology used for data extraction and analysis from ALICE data, aiding readers' understanding of the study's process.
There are sections where the complexity of the language might pose challenges for readers not deeply versed in the specific terminology. For instance, in the section discussing the "Clustering of Color Sources," concepts related to color strings, their interaction, and cluster formation might benefit from simplified explanations or additional context to aid broader comprehension.
Add brief descriptions or interpretations of key equations to help readers understand their significance without delving into the mathematical details.
Ensure that specialized terms are defined or explained, especially if they are not widely known. This is important for readers who may not be experts in the field.
Consider elaborating a bit more on the implications of the critical percolation density in the Color String Percolation Model (CSPM). How does it directly relate to the physical phenomena you're discussing?
When discussing the implications of your findings (such as the departure from LQCD results), provide more context. How significant are these deviations, and how might they impact current understandings or future research directions in high-energy physics?
Ensure that abbreviations like "DOF" (degrees of freedom) and Color String Percolation Model (CSPM) are defined upon their first use only. The former is repeatedly defined at line 24, 30, 228, 290 while the later one at lines 7, 74, 279.
Comments on the Quality of English Languageonly minor corrections are required
Author Response
Authors thank the referee for his constructive comments. Accordingly we have try to address all points.
- Section 2 has been renamed "Color String Percolation Model" to reflect the methodology used in the analysis.
- Line 77 explains the effect of color suppression factor.
- Discussion part has been expanded to include chiral condensate from LQCD and its implication. We have also include the future direction of our analysis.
- Duplicates have been removed.
Reviewer 2 Report
Comments and Suggestions for Authors
Report on universe-2782634
In this paper, experimental data from ALICE are combined with a mechanism of 'clustering of color sources'. This method, introduced by part of the authors some years ago (ref. [11]) rests on the percolation of the images of color strings projected to the transverse plane. As mechanism for the string breaking, the Schwinger method of pair creation known in QED_2 is used.
The paper gives a necessary introduction into the topic and explains the notions used. Also, it describes the stages in a heavy ions collision and several models, which are commonly in use. Relation to the lattice methods is also discussed. A basic observation is the relation between the multiplicity and the transverse momentum transfer. these are related to experimental data from the ALICE collaboration. Elaborating on these methods, the authors are able to obtain both, the temperature and the energy density of hot hadronic matter, from the experimental data. It is interesting to follow the estimation of the number of degrees of freedom (dof), following from energy and temperature in the high-T region. The results are compared with other approaches.
The paper touches an interesting and actual topic in high energy physics. It is well founded within the assumptions made, and it used the experimental data. As a result, a sound interpretation of the high energy ion collisions appears, which deepens our phenomenological understanding.
The relation between eq.(23), (24), and the number of DOF could be explained in some more detail.
Author Response
Authors thank the referee for his comments. Accordingly ,we have expanded the discussion of DOF. It has DOF from quarks and qluons separately to give total DOF.
Reviewer 3 Report
Comments and Suggestions for Authors
The manuscript represents a clearly written and very-well structured analysis dealing with the use of the Colour String Percolation Model (CSPM) to shed light on the initial stages in proton-proton, Xe-Xe, and Pb-Pb scatterings at LHC energies and kinematic cuts.
The article meets the criteria of quality, novelty and robustness for this journal.
I believe that the manuscript is suited for publication on MPDI Universe, modulo two minor corrections:
- Beginning of Section 2: Could you please elaborate more on the Schwinger QED_2 mechanism, possibly extending the quoted literature?
- For the sake of style, the Authors might want not to add acronyms, such as QGP and CSMP, in the Abstract. They could be easily inserted and explained in the Introduction.
Author Response
Authors thank the referee for his comments and suggestions.
- A part of QED_{2} is in Section 4 while dealing with temperature.
- We have taken care of acronyms . QGP and CSPM have been removed from the abstract.
Round 2
Reviewer 1 Report
Comments and Suggestions for Authors
The authors have responded to my comments, and based on their replies, I recommend the paper for publication in the journal "Universe".